# Effect of Interval between Human Chorionic Gonadotropin Priming and Ovum Pick-up on the Euploid Probabilities of Blastocyst

**DOI:** 10.3390/jcm9061685

**Published:** 2020-06-02

**Authors:** Chun-I Lee, Hsiu-Hui Chen, Chun-Chia Huang, Chien-Hong Chen, En-Hui Cheng, Jing Yang Huang, Maw-Sheng Lee, Tsung-Hsien Lee

**Affiliations:** 1Division of Infertility, Lee Women’s Hospital, Taichung 406, Taiwan; adoctor0402@gmail.com (C.-IL.); susan77kimo@yahoo.com.tw (H.-H.C.); agarhuang@gmail.com (C.-C.H.); clonemail@gmail.com (C.-H.C.); enhuicheng@gmail.com (E.-H.C.); msleephd@gmail.com (M.-S.L.); 2Institute of Medicine, Chung Shan Medical University, Taichung 402, Taiwan; 3Department of Obstetrics and Gynecology, Chung Shan Medical University Hospital, Taichung 402, Taiwan; 4Department of Medical Research, Chung Shan Medical University Hospital, Taichung 402, Taiwan; cshe282@csh.org.tw

**Keywords:** preimplantation genetic test, human chorionic gonadotropin priming, hCG–OPU interval, euploidy probability, blastocyst biopsy

## Abstract

This retrospective study attempts to elucidate the relevance of the interval between human chorionic gonadotropin priming and oocyte pick-up (hCG-OPU) to the euploidy probability of biopsied blastocysts in preimplantation genetic tests for aneuploidy (PGT-A) cycles. A total of 1889 blastocysts from 511 patients undergoing PGT- A cycles were used. An analysis of generalized estimating equations (GEE) was used to identify whether the hCG–OPU interval is associated with euploidy probabilities of blastocysts. Accordingly, maternal age (OR: 0.925, 95% CI: 0.903–0.948, *p* < 0.001) and the hCG–OPU interval (OR: 1.138, 95% CI: 1.028–1.260, *p* = 0.013) were the two significant factors associated with the euploidy probabilities. The Cochran-Armitage trend test demonstrated that the blastocyst euploidy percentage increased progressively with the increasing hCG-OPU interval in normal responders (*p* = 0.006) and advanced maternal age (age ≥38 years; *p* = 0.020) groups. In normal responders, the euploidy rate was highest in the 38–39 h interval (43.1%, 47/109). In contrast, the euploidy rate was lowest in the 34–35 h interval (28.7%, 29/105). In conclusion, the present study demonstrated that at an hCG-OPU interval between 34–39 h, the longer the hCG-OPU interval, the higher the probability of euploidy for blastocysts.

## 1. Introduction

Human chorionic gonadotropin (hCG) is normally administered as a substitute for the luteinizing hormone (LH) surge, and the interval from hCG administration to oocyte pick-up (OPU; i.e., hCG–OPU interval) is the period of oocyte maturation in vivo. Ovulation can occur anytime from 24–56 h after LH surge onset, with a mean time of 32 h [1]. In most in vitro fertilization (IVF) programs, the commonly used interval is 32–36 h [2]. Gudmundsson et al. [3] demonstrated that prolonging the interval to oocyte retrieval (extended by up to 39 h) can improve oocyte maturity without preoperative ovulation. Several studies of in vitro insemination cycles have suggested that extending the hCG–OPU interval improves oocyte maturation and increases fertilization rates [4,5]. Several studies have also suggested that prolonging the interval to oocyte retrieval increases the number of high-quality cleaving embryos in controlled ovarian hyperstimulation cycles [3,5]. Furthermore, an hCG–OPU interval of longer than 36 h has a predominant effect on the success of IVF [5,6,7,8] and in vitro maturation [9]. In addition, oocyte retrieval should not be attempted until 36 h after hCG injection [8].

Regarding IVF cycles, several reports have been published on the effect of extending the hCG–OPU interval to more than 36 h on oocyte maturation, embryonic developmental competence, and pregnancy rate [8]. The hCG–OPU interval is crucial for luteinization, the expansion of cumulus cells, and oocyte meiosis resumption [3]. During this interval, nuclear and cytoplasmic maturation of oocytes occurs in vivo, and oocytes acquire the ability to recommence meiotic maturation in response to gonadotropin stimulation. In this study, we determined whether extending the hCG–OPU interval affects the probability of euploidy when the maturation period is extended in vivo. Whether biopsied blastocysts from prolonged hCG–OPU intervals have a higher incidence of chromosomal normality compared with shorter intervals from preimplantation genetic tests for aneuploidy (PGT-A) cycles is unknown. This study thus retrospectively compares the incidence of euploidy in biopsied blastocysts derived from PGT-A cycles.

## 2. Materials and Methods

### 2.1. Patient Selection

This was a retrospective, single-center clinical trial to evaluate the effects of hCG–OPU interval on blastocyst euploidy rate. Figure 1 is a flowchart illustrating the study design. This study included 581 patients who had been referred to Lee Women’s Hospital, Taiwan and treated with PGT-A and frozen embryo transfer (FET) from July 2016 to December 2017. The criteria for PGT-A recommendation included (1) advanced age (≥38 years) or (2) young women (age < 38 years) with >1 prior IVF failure [10]. In addition, PGT-A applied to poor responders (i.e., patients with oocyte numbers below 5) or male factors such as severe oligoasthenoteratozoospermia (OAT), azoospermia and testicular/epididymal sperm aspiration or extraction (TESA/TESE) were grounds for exclusion from this study. All patients and their husbands signed standard IVF consent forms and underwent the same stimulation and FET protocols. All aneuploidy screenings were authorized by patients after consultation. All treatment histories and clinical outcomes of patients were recorded in the database system of Lee Women’s Hospital before analysis. The retrospective data analysis was approved by the Institutional Review Board of Chung Shan Medical University, Taichung, Taiwan (CS-18082).

### 2.2. Controlled Ovarian Stimulation and hCG–OPU Interval

Controlled ovarian stimulation, oocyte collection, and denudation were performed as described previously [11]. All patients were administered leuprolide acetate (Lupron, Takeda Chemical Industries, Ltd., Osaka, Japan), which was started during the midluteal phase for downregulation. All patients subsequently received recombinant follicle-stimulating hormone (Gonal-F; Serono, Bari, Italy) from cycle day 3 for ovarian stimulation until the dominant two follicles reached a diameter of >18 mm; this was followed by injection of 250 µg of hCG (Ovidriell, Serono) 34–39 h before oocyte retrieval. The hCG injection time for all patients was 7:00 p.m., but the time intervals between hCG and OPU varied as a result of laboratory workload on the OPU day. The order of the OPU was based on the registration order of the patient’s appointment on day 8 or 10 of the cycle, and the time of OPU was recorded. The interval between hCG and OPU was defined as the hCG–OPU interval, and the interval range was 34–39 h.

### 2.3. Insemination or Intracytoplasmic Sperm Injection Procedures and Embryo Culture

The retrieved oocytes were cultured in Quinn’s Advantage Fertilization Medium (Sage BioPharma, Inc., Trumbull, CT, USA) with 15% serum protein substitute (SPS, Sage BioPharma, Inc.) in a triple gas phase of 5% CO_2_, 5%O_2_, and 90% N_2_. After conventional insemination or intracytoplasmic sperm injection (ICSI), all embryos were further cultured in microdrops of a cleavage medium (Sage BioPharma, Inc.) with 15% SPS. The insemination or ICSI was performed 38–41 h after hCG injection. In the morning of day 3 (at 70–72 h after insemination or ICSI), all cleaved embryos were assessed and group cultured in microdrops of a blastocyst medium (Sage BioPharma, Inc.) with 15% SPS to culture to blastocyst for trophectoderm (TE) biopsy. Expanding and expanded blastocysts underwent TE biopsy on day 5 or 6. The blastocyst quality was assessed immediately before TE biopsy: the embryo evaluation and scoring were performed using the aforementioned systems [12], and only blastocysts considered to be of adequate quality (4, 5, 6, AB, BA, and BB [12]) were subjected to biopsy. To further analyze the effects of embryo quality on the euploidy, all biopsied blastocysts were further classified as either (1) excellent-quality (AA, AB) or (2) good-quality (BB, BA). After TE biopsy, blastocysts were cultured in microdrops of a blastocyst medium with 15% SPS and a three-gas incubator until vitrification.

### 2.4. TE Biopsy and Next-Generation Sequencing

Once the embryos reached the blastocyst stage, TE biopsy was performed as described by Chen et al. [11]. The biopsied TE cells were immediately placed in an RNAse–DNAse-free polymerase chain reaction tube and amplified using the SurePlex DNA Amplification System (Illumina, Inc., San Diego, CA, USA). The amplified products were analyzed through 1.5% agarose gel electrophoresis, and successfully amplified DNA had lengths in the range of 100 ± 1000 bp. Extracted cells were placed in 2 μL of buffer and shipped frozen to Genesis Genetics for PGT-A using a high-resolution, next generation sequencing (hr-NGS) platform.

### 2.5. Statistical Analysis

Our primary objective was to determine the effect of hCG–OPU intervals on euploid probability. The interrelations between the predictor variables were examined using Spearman’s rank correlation coefficients. According to the results of the correlation coefficients, the variables including the hCG-OPU interval, women age, male age, AMH, the oocyte number, ovarian response, total FSH dosage, blastocyst grade, the oocyte maturation rate, the fertilization rate and the blastocyst rate were included in a statistical analysis of the generalized estimating equation (GEE). The univariate GEE analysis was applied to evaluate the effect of a single factor related to the euploid probability. Multivariate relations between the predictor variables and the euploidy were statistically analyzed in GEE models [13]. Embryo euploidy (yes/no) served as outcome variables, as did predictors including hCG-OPU interval (per hour), women age (per year), total FSH dosage (per 1000 IU), ovarian response, the number of retrieved oocytes (per number), the oocyte maturation rate (per 10 percent), the fertilization rate (per 10 percent), the blastocyst rate (per 10 percent) and male age (per year). In order to facilitate the clinical application of hCG-OPU intervals, we further divided the hCG-OPU interval into 5 groups (i.e., 34–35, 35–36, 36–37, 37–38 and 38–39 h). Furthermore, all blastocysts were further divided into subgroups by age (<38 and ≥38 years) and ovarian response (normal and excessive responders) to further distinguish the effect of hCG-OPU intervals on the euploidy rates in these subgroups. It has been reported that the aneuploidy rate sharply rises in women ≥38 years [14]. Therefore, the biopsied blastocysts were classified by age as (1) <38 years and (2) ≥38 years [15]. The patients with >20 retrieved oocytes were defined as hyperresponders, based on the knowledge that the pregnancy rates did not increase when >20 oocytes were retrieved [16]. Patients with 5–20 oocytes were thus defined as normal responders.

We used standardized descriptive statistics, comparing categorical variables with the χ^2^ or Fisher’s exact test and continuous variables using the Student’s *t* test. A difference of <0.05 was considered significant. A Cochran-Armitage trend test was performed to look for trends between the hCG-OPU interval groups. All calculations were performed using SPSS (version 23.0; StatSoft Inc., Tulsa, OK, USA). The primary data used for this analysis are available as a Appendix A.

## 3. Results

### 3.1. The Study Design and Patient Characteristics

The study included 1889 blastocysts from 511 patients (cycles) which underwent TE biopsy. The clinical data and cycle parameters are detailed in Figure 1 and Table 1. The overall euploidy rate in the PGT-A cycles was 36.4% (687/1889).

### 3.2. Effect on Euploidy Probability of Women’s Ages and hCG–OPU Interval

According to univariate GEE analysis, the women’s ages (OR: 0.926, 95% CI: 0.907–0.945, *p* < 0.0001; Table 2) and men’s ages (OR: 0.980, 95% CI: 0.962–0.998, *p* = 0.031) were negatively associated with euploidy probability; but the hCG–OPU interval (OR: 1.133, 95% CI: 1.020–1.257, *p* = 0.019), oocyte number (OR: 1.013, 95% CI: 1.003–1.023, *p* = 0.010) and blastocyst rate (OR: 1.934, 95% CI: 1.067–3.507, *p* = 0.030; Table 2) were positively associated with the euploidy probability. According to a multivariate GEE analysis of several key factors, only the women’s age (OR: 0.925, 95% CI: 0.903–0.948, *p* < 0.0001) and the hCG–OPU interval (OR: 1.138, 95% CI: 1.028–1.260, *p* = 0.013) were the significant predictors of blastocyst euploidy probability; the oocyte number, blastocyst rate and male age were not associated with the blastocyst euploidy probability.

### 3.3. Effects of the hCG–OPU Interval on Euploidy Rates in Different Subgroups

All biopsied blastocysts were divided into five groups according to the hCG–OPU interval length. The data exhibited a generally significant upward trend in the euploidy probability as the hCG–OPU interval increased (*p* = 0.019, Table 3). In normal responders, a significantly upward trend revealed by the Cochran-Armitage trend test in euploidy rate occurred as the hCG–OPU interval increased (*p* = 0.006). The euploidy rate was highest in the 38–39 h interval (43.1%, 47/109; Table 3). In contrast, the euploidy rate was lowest in the 34–35 h interval (28.7%, 29/101). In excessive responders, a nonsignificant upward trend was revealed by a Cochran-Armitage trend test in the euploidy rate as the hCG–OPU interval increased. However, the euploidy rate in the 34–35 h interval (26.2%, 16/61) was lower than those in the other interval groups.

In the group of women aged <38 years, a nonsignificant upward trend was revealed by a Cochran-Armitage trend test in euploidy rate as the hCG–OPU interval increased. However, the Cochran-Armitage trend test demonstrated that the the blastocyst euploidy percentage increased progressively with the increasing hCG-OPU interval (*p* = 0.020) in the older age (age ≥38 years) group. The euploidy rate for the group with an interval of 34–35 h (13.2%, 5/38) was lowest, and that for the group with an interval of 38–39 h (24.1%, 13/54) was highest.

## 4. Discussion

In this study, the primary factors associated with blastocyst euploidy rate were shown to be women’s ages and the hCG–OPU interval, in a logistic regression analysis using the GEE model. With advancing age, the quantity and quality of oocytes deteriorate [17]. Meiotic abnormalities arising in oocyte increase in frequency by 10–60%, or even more, with advanced maternal age [18]. A higher risk of chromosomal abnormalities in oocytes from women of advanced age further leads to increased embryonic aneuploidy [19]. However, in addition to the female age factor, we found a general upward trend in euploidy probability as the hCG–OPU interval increased. The results indicate that the probability of euploidy blastocysts can be increased by prolonging the hCG–OPU interval from 34–39 h, resulting in a higher euploidy probability of biopsied blastocyst.

The hCG–OPU interval is critical for the start of luteinization, expansion of cumulus cells, and the resumption of oocyte meiosis [3]. In contrast, errors in the meiosis processes, including deficient spindle structures and the unbalanced allocation of chromosomes, could lead to aneuploidy [20,21]. Because the hours after luteinizing stimulus represent a period of intense nuclear and cytoplasmic activity in human oocytes, the interval between hCG priming and oocyte retrieval potentially determines the degree of cellular and cytogenetic maturation. Nuclear and cytoplasmic oocyte maturation is responsible for fertilization and high-quality embryo development. Although nuclear maturity and completion of the first meiotic division are readily achieved in vitro subsequent to the presence of the first polar body, the processes of the second meiosis and cytoplasmic maturation still need time to reach completion. Furthermore, the early removal of the cumulus cells reduces the chances of the process properly reaching cytoplasmic maturity [22]. Oocyte cytoplasmic immaturity is related to metaphase plate anomalies and aneuploidies [23]. Consequently, the competence of the human oocyte cannot be precisely predicted only by observation of the follicle size or the presence of the first polar body at the time of insemination or injection [24].

The prolonged hCG–OPU interval increased the production of oocytes with fully expanded cumuli, with the assumption that the longest interval would yield oocytes in which all cellular and nuclear maturation processes would have been completed, and the mature oocytes collected would therefore be more likely to develop into normally cleaving embryos [5]. In addition, many factors involved in oocyte maturation and embryo development, such as angiotensin II, vascular endothelial growth factor, interleukin (IL)1, IL-6, IL-8, angiopoietin, insulin-like growth factor, basic fibroblast growth factor, and endothelin, have a time-dependent effect after hCG priming [4,25,26]. Elongation of the hCG–OPU interval, oocyte maturation, fertilization, and cleavage rates were significantly higher in oocytes retrieved >36 h after hCG administration than in those retrieved <36 h after hCG administration [3,5]. Indeed, we observed in the present study that a longer hCG–OPU interval (38–39 h) was associated with a higher euploidy rate and a shorter hCG–OPU interval (34–35 h) with a lower euploidy rate. Thus, we suggest that the elongation of the hCG–OPU interval may improve meiosis completion and oocyte maturation in vivo to increase the euploidy blastocyst probability. 

In the present study, prolonging the hCG–OPU interval increased the blastocyst euploidy probability, especially in the advanced maternal age (≥38 years) group and in normal responders. Nonetheless, the short hCG-OPU interval (34–35 h) is still associated with the lowest euploidy rates of biopsied blastocyst compared to those of longer hCG-OPU intervals in the young age (<38 years) or the excessive responders. In normal responders, the 38–39-h interval yielded the optimal euploidy rate. However, in the excessive responder group, a nonsignificant upward trend was revealed by the Cochran-Armitage trend test in euploidy rate as the hCG–OPU interval increased. We suggest that the hCG action in euploidy, even for different ovarian responses, is still influential. The period of hCG effect on oocyte maturation may be limited in excessive responders. Although the underlying causes are still unclear, the patients with polycystic ovary syndrome (PCOS) were not excluded in this study; hyperresponse to controlled ovarian stimulation is known to be a common characteristic for PCOS patients [27]. It is therefore reasonable to conclude that a portion of the excessive responders in this study might be diagnosed with PCOS. Geng et al. [28] reported that 21% of PCOS patients show premature luteinization, especially before follicle maturation, suggesting poor oocyte quality in PCOS patients [29]. Additionally, the metabolic similarities observed in the follicle fluid between excessive responders and PCOS patients indicate that metabolic alteration, among other things, could impair the quality of at least a portion of oocytes and embryos in excessive responders [27]. Therefore, a proportion of premature luteinization and metabolic abnormalities in excessive responders might limit the effect of prolonging the hCG-OPU interval on improving the euploidy rate.

The oocyte number was not associated with euploidy probability in the present study after multivariate logistic regression analysis. Although the oocyte number is positively related to the euploid probability in univariate regression analyses, the consideration of multiple factors demonstrated that only maternal age and hCG-OPU intervals were distinct predictors for embryo euploidy, which reached statistical significance in the multivariate regression analysis. Similarly, Kahraman [30] demonstrated that the euploidy rate does not directly correlate with the retrieved oocyte number, and that female age is the major predictor of euploidy rate. Morin et al. [31] also demonstrated that postcycle oocyte yield is not associated with a high aneuploidy rate in patients < 38 years.

Blastocyst quality was not associated with euploidy probability in the present study. Capalbo et al. [32] reported that blastocyst morphology is weakly predictive of embryo’s ploidy status in a comprehensive chromosome screening result, indicating that the euploidy rate appears elevated in blastocysts with high quality morphology. However, in this study, only the good quality blastocysts were collected for TE biopsy and PGT-A, suggesting that the effect of embryo quality on euploid probability might be minimized. Notably, this study using hr-NGS for PGT-A differed from previous data using array comparative genomic hybridization analyses, which might be another reason for the different outcomes [33].

The male infertility factor has been found to increase the risk of aneuploidy following fertilization [34,35]. However, patients with severe OAT and surgical sperm retrieval were excluded in this study, and the effect of sperm quality was thus not discussed. Previous reports have stated that neither male age nor semen parameters influence clinical pregnancy or live birth outcomes of IVF [36], and that meiotic errors are rare in sperm cells [37]. In this study, the male age was not associated with euploidy probability after multivariate regression analysis with the GEE model; we suggest that the meiotic error from sperm is not a major factor related to the effect of the hCG–OPU interval on euploidy probability.

Although the extension of the hCG–OPU interval is beneficial, the cancellation of treatment cycles as a result of spontaneous ovulation must be avoided. Andersen et al. [38] noted a mean time interval of 38.3 h from hCG injection to first follicular rupture. However, several studies have demonstrated that no women ovulated up to 39 h from hCG administration [3,39,40]. In our study, no women ovulated even up to 39 h after hCG injection.

## 5. Conclusions

Our findings indicate that the hCG–OPU interval in PGT-A is associated with the euploidy rate, especially in women of advanced maternal age (≥38 years) and normal responders. As expected, female age is strongly correlated to euploidy rate in PGT-A cycles. Nonetheless, human chorionic gonadotropic priming and oocyte pick-up intervals toward the upper limit of 34–39 h are associated with a higher chance of euploidy for blastocysts. A general upward trend in euploidy probability occurred as the hCG–OPU interval increased. Therefore, we consider that prolonging the hCG–OPU interval in PGT-A cycles is a valuable strategy to collect more euploidy blastocysts. In addition, for normal responders, the hCG–OPU interval of 38–39 h was associated with a significantly increased blastocyst euploidy rate. In contrast, the 34–35 h interval demonstrated an adverse effect on the euploidy rates of biopsied blastocysts.

## Figures and Tables

**Figure 1 jcm-09-01685-f001:**
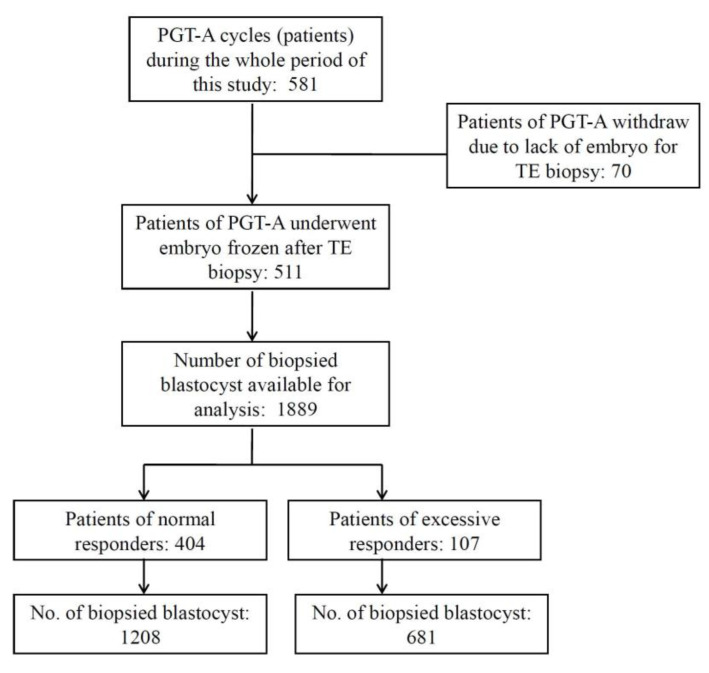
Study design. This study included 581 preimplantation genetic tests for the aneuploidy (PGT-A) cycles of infertile couples. There were 1889 blastocysts from 511 cycles, which underwent trophectoderm (TE) biopsy and PGT-A. All biopsied blastocysts were divided into five groups according to the hCG–OPU interval length. Only expanded blastocysts (grades 4, 5, and 6) with qualified inner cell mass and trophectoderm morphology (grades AB, BA, and BB) were subjected to TE biopsy.

**Table 1 jcm-09-01685-t001:** Baseline characteristics of women included in this study.

Baseline Characteristics	Normal Responders	Excessive Responders	Total Patients
Patients	404	107	511
Age (year)	37.4 ± 5.0 ^a^	33.4 ± 5.5 ^a^	36.4 ± 5.4
AMH (ng/mL)	3.5 ± 2.8 ^b^	8.3 ± 4.2 ^b^	4.6 ± 3.7
No of cycles in each group of hCG-OPU interval			
34–35 h	31	10	40
35–36 h	96	17	104
36–37 h	139	36	171
37–38 h	105	33	135
38–39 h	33	11	42
Mean of hCG-OPU interval	36.3 ± 1.0	36.4 ± 1.1	36.3 ± 1.0
Total gonadotropin dose	3350 ± 718 ^c^	3101 ± 624 ^c^	3290 ± 702
No. of oocytes	11.8 ± 5.0 ^d^	29.8 ± 7.4 ^d^	15.8 ± 9.3
Oocyte maturation rates (mean ± SD)	78.7 ± 13.0	80.8 ± 11.6	79.2 ± 12.8
Fertilization rates (mean ± SD)	78.5 ± 19.5 ^e^	74.0 ± 16.1 ^e^	77.3 ± 19.0
Blastocyst rates (mean ± SD)	46.0 ± 23.4 ^f^	40.3 ± 17.9 ^f^	44.6 ± 22.5
Total biopsied blastocyst no.	1208	681	1889
Blastocyst morphology			
Excellent quality (%)	14.2 (172/1208)	15.7 (107/681)	14.8 (279/1889)
Good quality (%)	85.8 (1036/1208)	84.3 (574/681)	85.2 (1610/1889)
Euploidy rate (%)	34.9 (421/1208)	39.1 (266/681)	36.4 (687/1889)

The ovarian response groups were classified as (1) normal responders (5–20 oocytes), or (2) excessive responders (>20 oocytes). Embryo morphology grades were defined as follows: excellent-quality blastocysts (inner cell mass and trophectoderm grade is AA or AB) and good-quality blastocysts (inner cell mass and trophectoderm grade is BB or BA); ET: embryo transfer. χ2 was used to compare the categorical variables; a t test was used to compare the continuous variables; a,b,d - *p* < 0.001, c - *p* = 0.002, e - *p* = 0.023, f - *p* = 0.010.

**Table 2 jcm-09-01685-t002:** Univariate and Multivariate GEE analysis of factors related to the euploid probability of a biopsied blastocyst.

**Univariate Regression Analysis**				
Variables	**B**	**Odds Ratio**	**95% CI ***	***p* Value**
hCG-OPU interval, per 1 increment (h)	0.125	1.133	1.020–1.257	0.019
Women age, per 1 increment (yr)	−0.077	0.926	0.907–0.945	<0.0001
AMH	0.017	1.017	0.987–1.047	0.265
Oocyte numbers, per 1 increment	0.013	1.013	1.003–1.023	0.010
Ovarian response groups, excessive v.s. normal	0.185	0.831	0.651–1.060	0.136
Embryo grades, A v.s. B	−0.083	0.921	0.843–1.006	0.067
Total FSH dosage, per 1,000 increment (IU)	0.000	1.000	1.000–1.000	0.105
Maturation rates, per 10 increment (%)	0.665	1.945	0.717–5.273	0.191
Fertilization rates, per 10 increment (%)	0.187	1.206	0.610–2.383	0.590
Blastocyst rates, per 10 increment (%)	0.660	1.934	1.067–3.507	0.030
Male age, per 1 increment (yr)	−0.020	0.980	0.962–0.998	0.031
**Multivariate Regression Analysis**				
Variables	**B**	**Odds Ratio**	**95% CI ***	***p* Value**
hCG-OPU interval, per 1 increment (h)	0.129	1.138	1.028–1.260	0.013
Women age, per 1 increment (yr)	−0.078	0.925	0.903–0.948	<0.0001
Oocyte numbers, per 1 increment	−0.006	0.995	0.983–1.006	0.361
Blastocyst rates, per 10 increment (%)	0.357	1.429	0.743–2.750	0.284
Male age, per 1 increment (yr)	−0.004	0.996	0.979–1.013	0.653

* CI: confidence intervals. Statistical analysis was performed using the logistic regression analysis for the clustered nature of the data and multivariate regression models to adjust for potential confounders. The oocyte maturation rate = MII oocyte number/the retrieved oocyte number; the fertilization rate = the 2PN number/the MII oocyte number; blastocyst rate = blastocyst number/2PN number.

**Table 3 jcm-09-01685-t003:** Comparison of euploidy outcomes between different hCG–OPU intervals in different patient subgroups.

Groups	Euploidy Rates (%)					Cochran-Armitage Trend Test
hCG-OPU Intervals	34–35 h	35–36 h	36–37 h	37–38 h	38–39 h	*p* Values
Overall	27.8 (45/162)	35.2 (144/409)	39.9 (241/604)	37.4 (187/500)	40.9 (70/171)	0.019
Normal responders	28.7 (29/101)	31.4 (88/280)	34.1 (143/419)	38.1 (114/299)	43.1 (47/109)	0.006
Excessive responders	26.2 (16/61)	43.3 (56/129)	43.0 (98/228)	36.3 (73/201)	37.1 (23/62)	0.880
Women age < 38 years	32.3 (40/124)	43.3 (109/252)	43.1 (195/452)	39.8 (127/319)	48.7 (57/117)	0.122
Women age ≥ 38 years	13.2 (5/38)	22.3 (35/157)	25.1 (46/195)	23.1 (60/181)	24.1 (13/54)	0.020

A Cochran-Armitage trend test was performed to look for trends between the hCG-OPU interval groups. The ovarian response groups were classified as (1) normal responders (5–20 oocytes) or (2) excessive responders (>20 oocytes).

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
