# Peer review of "Effect of Interval between Human Chorionic Gonadotropin Priming and Ovum Pick-up on the Euploid Probabilities of Blastocyst"

_jcm, 2020, doi:10.3390/jcm9061685_

Round 1
Reviewer 1 Report
This paper will be very interesting for your readers.
Please remove the first 8 lines of the introduction (35-42).
Although the elongation of hCG–OPU interval seems to improve the euploid probabilities of blastocyst, please try to better explain why this result was observed in this particular time interval. In other words, which is exactly the theoretical ground of having the best results on 38-39h hCG–OPU interval (and not after 39h or 40h). Your comment related to the increased age of the women is confirmed in epidemiological data. However, there are no “unexpected” findings in these observations. A comment about the expected results related to age would emphasize this understanding. Furthermore, the phrase “Human chorionic gonadotropic priming and oocyte pick-up intervals towards the upper limit of 34-39 h are associated with a higher euploidy chance for blastocysts” summarizes your findings and it should be included in the conclusion after the discussion. After the enrichment of the text with these comments, the paper could be published.
Author Response
Thank you for the precious comments.
- We remove the first 8 lines of the introduction.
- We add the following paragraph into the introduction,
“Regarding IVF cycles, several reports have been published on the effect of extending the hCG–OPU interval to more than 36 h on oocyte maturation, embryonic developmental competence, and pregnancy rate [8]. The hCG–OPU interval is crucial for luteinization, expansion of cumulus cells, and oocyte meiosis resumption [3]. During this interval, nuclear and cytoplasmic maturation of oocytes occurs in vivo, and oocytes acquire the ability to recommence meiotic maturation in response to gonadotropins stimulation. In this study, we determined whether extending the hCG–OPU interval affects the euploidy probability when the maturation period is extended in vivo. Whether biopsied blastocysts from prolonged hCG–OPU intervals have a higher incidence of chromosomal normality as compared with shorter intervals from preimplantation genetic tests for aneuploidy (PGT-A) cycles is unknown. This study thus retrospectively compared the incidence of euploidy in biopsied blastocysts derived from PGT-A cycles.”
Furthermore, the last paragraph in the Discussion explained why we choose the time interval limited within 34-39 hours. Due to the risk of pre-operative ovulation for a hCG-OPU interval >39h, we tried to avoid such condition in the daily practice and, consequently, extremely rare cycles featured a hCG-OPU interval >39h in our center. - We add the sentence “As expected, female age is a major factor correlated to euploidy rate in PGT-A cycles. Nonetheless, human chorionic gonadotropic priming and oocyte pick-up intervals towards the upper limit of 34-39 h are associated with a higher euploidy chance for blastocysts.” into the section of Conclusion.
Reviewer 2 Report
In the current study, the authors showed that hCG–OPU interval in PGT-A was associated with the euploidy rate. This effect was more pronounced in advanced maternal age and normal responders. The study is of significant importance and performed and presented well, however, it needs some minor points to be addressed.
- The starting paragraph of the introduction contains the original text of the journal instruction which needs to be removed.
- The authors discussed the role of polycystic ovary syndrome (PCOS) on hyper response to controlled ovarian stimulation which may also be correlated as the effect of increased testosterone (a common occurrence in PCOS patients). Is there any effect of fetal sex on euploidy rates? This may be an important finding corroborating with the PCOS condition.
Author Response
Response:
Thank you for the precious comments.
First, the original test of the journal instruction in the starting paragraph is removed.
Second, the hyper-response in PCOS patients may be correlated to the effect of increased testosterone. The effect of fetal sex on euploidy rates is an interesting point. Nonetheless, according to our data of biopsied blastocyst (the following table), the fetal sex does not correlate to euploidy rates.
Baseline characteristics |
Normal responders |
Excessive responders |
Total patients |
Sex ratio of biopsied blastocyst |
|
|
|
Male (%) |
50.7 (613/1208) |
51.1 (348/681) |
50.9 (961/1889) |
Female (%) |
49.3 (595/1208) |
48.9 (333/681) |
49.1 (928/1889) |
Euploidy rate of fetal sex |
|
|
|
Male (%) |
36.2 (222/613) |
38.5 (134/348) |
37.0 (279/961) |
Female (%) |
33.5 (199/594) |
39.6 (132/333) |
35.7 (331/928) |